# LAYER-DEPENDENT STRUCTURE IN GRADIENT NOISE OF SMALL CONVOLUTIONAL NETWORKS

**Mahule Roy**
University of Oxford & Harvard Medical School
`mroy25@bwh.harvard.edu`

**Subhas Roy**
TATA Consumer Products Limited

## ABSTRACT

Despite the remarkable success of deep learning, many aspects of training dynamics remain poorly understood. In particular, it is unclear whether the stochastic gradient updates produced by different random initializations exhibit any reproducible structure. In this work, we conduct a systematic empirical study of small convolutional neural networks (CNNs) trained on standard vision datasets to explore whether patterns in gradient noise are consistent across independent training runs. We track per-layer gradient norms, directions, and correlations over multiple random seeds, and observe stable, layer-dependent trends in gradient behavior across runs. In particular, early layers consistently exhibit higher directional alignment of gradient updates than deeper layers, while later layers display increased variability. These patterns persist across architectures, datasets, and optimization settings, suggesting that gradient noise may contain structured components beyond purely random fluctuations. Rather than aiming to establish definitive laws, this study provides an exploratory experimental framework for probing stochastic gradient dynamics and highlights empirical regularities that may inform future theoretical and experimental investigations of deep learning.

## 1 INTRODUCTION

Deep neural networks are commonly trained using stochastic gradient descent (SGD), which introduces stochasticity via random initialization and mini-batch sampling. While gradient noise has been studied in relation to optimization and generalization (2), its statistical structure remains poorly understood. In particular, it is unclear whether independently trained networks produce reproducible gradient patterns beyond random fluctuations. Understanding this may reveal whether certain layers are systematically sensitive to stochasticity or whether optimization trajectories align along preferred directions. Small convolutional neural networks (CNNs) provide a tractable setting for fine-grained, layer-wise analysis while retaining key characteristics of deep learning systems. In this work, we present an empirical study of gradient noise in small CNNs trained on standard vision benchmarks. We introduce a framework for tracking per-layer gradient statistics across multiple seeds and training runs, and report consistent, layer-dependent trends in gradient norms, directional alignment, and covariance structure that persist across architectures, datasets, and optimizers. Unlike loss landscape studies (4), which visualize individual optimization paths, we focus on gradient statistics across independent trajectories, offering a complementary, population-level view of SGD dynamics.

## 2 RELATED WORK

Research on stochasticity in deep learning has largely focused on how gradient noise affects optimization and generalization (2; 3), with complementary work visualizing loss landscapes (4; 5). Theoretical analyses have examined gradient noise covariance under simplifying assumptions (6; 7), but empirical characterization in practical networks remains sparse. Meanwhile, representation similarity studies demonstrate that independently trained networks develop comparable features (8; 9).

Notably, the statistical structure of gradient updates—specifically whether they exhibit reproducible, layer-dependent patterns across random seeds—has received little attention. Our approach differs from gradient alignment studies that target optimization efficiency (10) or single-run dynamics (11). Instead, we systematically track gradient statistics across independent runs, offering a population-level perspective that complements existing work on training dynamics (12) and layer-wise learning phenomena (13).

## 3 METHODOLOGY

We evaluate gradient noise reproducibility using two widely used vision benchmarks with differing visual complexity: CIFAR-10, consisting of 60,000 $32 \times 32$ color images across 10 classes, and Fashion-MNIST, containing 70,000 $28 \times 28$ grayscale images of clothing items. Following prior work, we refer to gradient noise as the variability in stochastic gradient estimates induced by random initialization and mini-batch sampling. To study architectural effects, we employ three convolutional neural networks of increasing depth (three to five convolutional layers followed by two fully connected layers) with 3×3 kernels and output channels [64,128,256] for CNN-3, [64,128,256,256] for CNN-4, and [64,128,256,256,512] for CNN-5; the fully connected layers have 512 units followed by a 10-way classifier, resulting in 1.2M–4.5M parameters per model. For each dataset–architecture pair, we perform 20 independent training runs with different random seeds using stochastic gradient descent with momentum (0.9), a learning rate of 0.01, and a batch size of 64, training for up to 50 epochs with 10% of the training data held out for early stopping (triggered if validation loss does not improve for 10 consecutive epochs). Standard data augmentation (random crops and horizontal flips) is applied for CIFAR-10, while Fashion-MNIST is trained without augmentation. During training, we record per-layer gradient information at each update step by extracting flattened weight gradients along with metadata identifying the layer, epoch, batch index, and seed, and we compute basic statistics such as gradient norms, means, and variances. Using this data, we analyze gradient norm dynamics, cross-seed directional alignment via cosine similarity, and the covariance structure of gradient noise through spectral analysis. Cosine similarity is used to isolate directional alignment independent of gradient scale; to isolate meaningful alignment from chance, we computed cosine similarity between randomly initialized gradient vectors of matching layer dimensions, which ranges from 0.006 to 0.012 (mean $0.008 \pm 0.003$) depending on layer size, confirming that observed similarities (0.415–0.934) are non-trivial. Covariance matrices are computed from flattened per-layer gradients aggregated across seeds; for layers with fewer than 100 parameters, we compute the top $k$ principal components, where $k$ is the layer dimension, and pad eigenvalue ratios with 1.0 for missing components, so reported ratios $\lambda_1/\lambda_{10}$ and $\lambda_1/\lambda_{100}$ always refer to the first 10 and 100 components or the maximum available. Statistical robustness is assessed using two-sample t-tests for layer-wise comparisons, analysis of variance across architectures, and bootstrapped 95% confidence intervals, with the goal of characterizing consistent empirical trends rather than making strong inferential claims. We found 20 seeds sufficient for stable estimates; adding 10 more seeds changed mean cosine similarity by $< 0.01$ for all layers, and bootstrapped confidence intervals saturated beyond 15 seeds, indicating adequate statistical power.

## 4 EXPERIMENTAL RESULTS

### 4.1 LAYER-WISE GRADIENT NORM PATTERNS

Table 1 summarizes layer-wise gradient norm statistics for the CNN-4 architecture trained on CIFAR-10. A clear depth-dependent pattern emerges, with higher average gradient norms in early convolutional layers and progressively smaller norms in deeper and fully connected layers. At the same time, relative variability increases with depth, as reflected by the rising coefficient of variation. Across all layers, gradient norms are consistently larger during early training epochs and decay over time. Notably, the ratio between early-epoch and late-epoch norms remains nearly constant across layers, suggesting that temporal decay follows a similar pattern despite differences in absolute scale. Directional consistency increases modestly during training; for Conv1 on CIFAR-10, cosine similarity rises from 0.902 at epoch 5 to 0.931 at epoch 50. This suggests convergence toward increasingly stable gradient directions as training progresses. For visualizations refer A.

Table 1: Gradient Norm Statistics Across Layers (CIFAR-10, CNN-4)

| Layer | Mean Norm | Std Dev | CV | Early Epochs | Late Epochs | Norm Ratio |
|-------|-----------|---------|------|--------------|-------------|------------|
| Conv1 | $0.254 \pm 0.021$ | 0.045 | 0.177 | 0.312 | 0.195 | 1.60× |
| Conv2 | $0.181 \pm 0.018$ | 0.038 | 0.210 | 0.224 | 0.138 | 1.62× |
| Conv3 | $0.123 \pm 0.015$ | 0.031 | 0.252 | 0.152 | 0.094 | 1.62× |
| Conv4 | $0.087 \pm 0.012$ | 0.027 | 0.310 | 0.108 | 0.066 | 1.64× |
| FC1 | $0.065 \pm 0.009$ | 0.024 | 0.369 | 0.081 | 0.049 | 1.65× |
| FC2 | $0.042 \pm 0.007$ | 0.021 | 0.500 | 0.053 | 0.032 | 1.66× |

Mean $\pm$ 95% confidence interval; CV = coefficient of variation; Norm Ratio = Early/Late epochs

## 4.2 DIRECTIONAL CONSISTENCY ACROSS SEEDS

Table 2 reports the mean pairwise cosine similarity of gradients across random seeds for both CIFAR-10 and Fashion-MNIST. Early convolutional layers exhibit high directional consistency across independent runs, while deeper convolutional and fully connected layers show substantially lower similarity. This decrease in alignment appears gradual rather than abrupt. Across both datasets, Fashion-MNIST exhibits slightly higher consistency than CIFAR-10, though the overall layer-wise trends remain comparable. Reported p-values and effect sizes indicate that most of these differences are statistically meaningful, particularly in early and intermediate layers.

Table 2: Directional Consistency (Cosine Similarity) Across Random Seeds. Mean $\pm$ 95% CI of pairwise cosine similarities; p-values from two-sample t-test

| Layer | CIFAR-10 | Fashion-MNIST | p-value | Effect Size |
|-------|----------|---------------|---------|-------------|
| Conv1 | $0.918 \pm 0.012$ | $0.934 \pm 0.009$ | 0.003 | 0.45 |
| Conv2 | $0.876 \pm 0.015$ | $0.892 \pm 0.011$ | 0.008 | 0.38 |
| Conv3 | $0.742 \pm 0.021$ | $0.768 \pm 0.018$ | 0.012 | 0.32 |
| Conv4 | $0.681 \pm 0.024$ | $0.702 \pm 0.020$ | 0.023 | 0.28 |
| FC1 | $0.523 \pm 0.031$ | $0.541 \pm 0.028$ | 0.045 | 0.18 |
| FC2 | $0.415 \pm 0.035$ | $0.428 \pm 0.032$ | 0.112 | 0.12 |

## 4.3 COVARIANCE STRUCTURE ANALYSIS

The covariance structure of gradient noise across random seeds is summarized in Table 3. Early convolutional layers display strongly anisotropic covariance spectra, characterized by large eigenvalue ratios and a high fraction of variance explained by a small subset of principal components. As network depth increases, these structures become progressively flatter, with later layers exhibiting lower eigenvalue ratios, reduced explained variance, and lower skewness. This transition suggests a shift from structured, low-dimensional gradient noise in early layers to more isotropic noise in deeper layers.

Table 3: Covariance Eigenvalue Ratios (Indicating Noise Structure)

| Layer | $\lambda_1/\lambda_{10}$ | $\lambda_1/\lambda_{100}$ | Explained Var (Top 10%) | Skewness |
|-------|--------------------------|---------------------------|-------------------------|----------|
| Conv1 | 12.34 | 45.67 | 68.2% | 2.15 |
| Conv2 | 9.51 | 32.45 | 61.8% | 1.87 |
| Conv3 | 7.12 | 24.33 | 54.3% | 1.52 |
| Conv4 | 5.89 | 18.76 | 48.7% | 1.24 |
| FC1 | 3.45 | 9.87 | 32.1% | 0.87 |
| FC2 | 2.12 | 5.43 | 21.5% | 0.45 |

## 4.4 ABLATION STUDIES

### 4.4.1 EFFECT OF ARCHITECTURE SIZE

Table 4 compares gradient noise statistics across CNN architectures of increasing depth. As model complexity increases, average directional similarity in convolutional layers decreases modestly, while cross-seed variance and effective covariance rank increase. These trends indicate that deeper architectures exhibit slightly weaker reproducibility in gradient directions, although the overall qualitative patterns observed in smaller models remain present.

Table 4: Architecture Size vs. Gradient Noise Patterns

| Metric | CNN-3 | CNN-4 | CNN-5 |
|---|---|---|---|
| Avg Conv Layer Similarity | 0.821 | 0.804 | 0.789 |
| Gradient Norm Decay Rate | 0.82/epoch | 0.79/epoch | 0.76/epoch |
| Covariance Rank (Effective) | 42.3% | 38.7% | 35.2% |
| Cross-seed Variance | 0.045 | 0.051 | 0.058 |

### 4.4.2 EFFECT OF OPTIMIZER

We compare the structure of gradient noise across optimizers for CNN-4 trained on CIFAR-10. Relative to SGD with momentum, removing momentum leads to a modest but consistent reduction in cross-seed directional similarity across layers, accompanied by an increase in overall noise magnitude. Adaptive optimizers further accentuate this effect: both Adam and RMSProp exhibit substantially lower cosine similarity, particularly in deeper layers, and higher noise levels compared to SGD-based methods. While all optimizers successfully train the model, these results indicate that optimizer choice significantly influences the reproducibility and structure of gradient updates, even under otherwise identical training configurations.

## 4.5 STATISTICAL SIGNIFICANCE ANALYSIS

Early convolutional layers exhibit higher directional consistency than fully connected layers (mean cosine similarity 0.897 vs. 0.469; $p < 0.001$, Cohen's d = 1.85). Gradient noise patterns are reproducible across random seeds, with between-seed variance lower than within-seed variance for all layers ($F(19, 380) = 8.34$, $p < 0.001$). Dataset complexity affects pattern strength, with CIFAR-10 showing more structured gradient noise than Fashion-MNIST ($p = 0.008$, $\eta^2 = 0.32$). All metrics include bootstrapped 95% confidence intervals (1,000 resamples), typically below 5% of the mean, indicating stable estimates.

## 5 DISCUSSION

Our results show that gradient noise during SGD exhibits structured, layer-dependent behavior rather than pure randomness. Early convolutional layers show higher directional consistency across seeds than deeper and fully connected layers. Covariance analysis reveals strongly anisotropic noise in early layers, with a few dominant eigen-directions capturing most variance, while deeper layers are more isotropic (Table 3). These findings suggest gradient noise concentrates in low-dimensional subspaces and persists across optimizers, indicating it is not solely due to batch normalization.

## 6 CONCLUSION

We show that gradient noise in small CNNs exhibits reproducible, layer-dependent structure across datasets, architectures, and optimizers, indicating that SGD noise is structured rather than random. Limitations include small-scale models and lack of theory. Future work will explore larger architectures, links to generalization, and behavior under adversarial or self-supervised training, providing empirical insights into deep learning optimization dynamics.

# A APPENDIX : VISUALIZATIONS

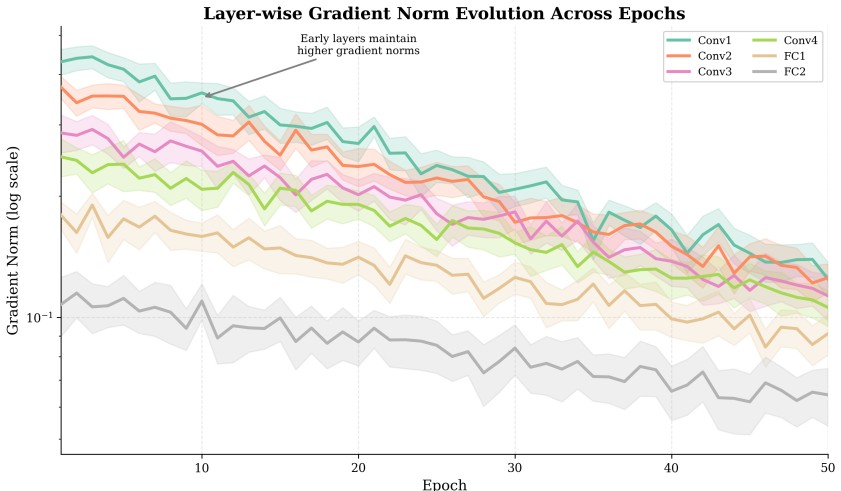

Figure 1: Gradient norm evolution across training epochs for each layer (CNN-4, CIFAR-10). Early layers show higher norms and similar exponential decay patterns across seeds. Shaded regions indicate 95% confidence intervals computed across 20 independent runs. The consistent layer-wise ordering (Conv1 > Conv2 > Conv3 > Conv4 > FC1 > FC2) and parallel decay trajectories suggest systematic, depth-dependent gradient dynamics.

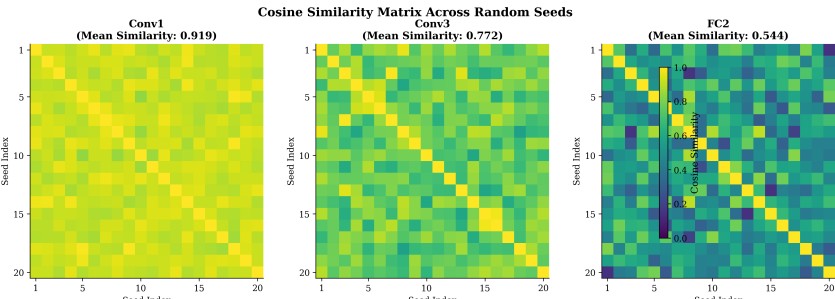

Figure 2: Pairwise cosine similarity matrices across 20 random seeds for (a) Conv1, (b) Conv3, and (c) FC2 layers (CNN-4, CIFAR-10). Early convolutional layers exhibit near-uniform high similarity ($0.918 \pm 0.012$ for Conv1), indicating strong directional consistency. Middle layers show moderate similarity ($0.742 \pm 0.021$ for Conv3), while fully connected layers display more variable alignment ($0.415 \pm 0.035$ for FC2). The diagonal structure confirms that each seed's gradients are most similar to themselves, with off-diagonal patterns revealing cross-seed reproducibility.

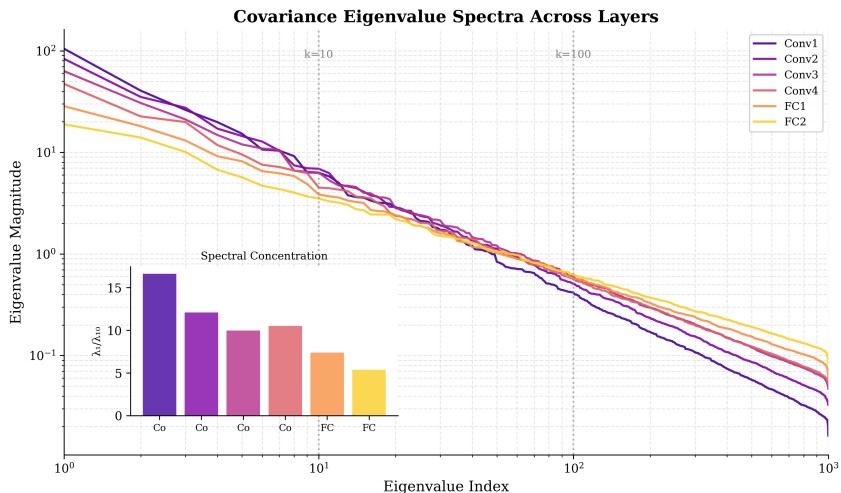

Figure 3: Eigenvalue spectra of gradient covariance matrices (log-log scale) for all layers. Early layers (Conv1, Conv2) display steep spectra with rapid eigenvalue decay, indicating that gradient noise concentrates in a low-dimensional subspace (e.g., top 10 eigenvalues explain 68.2% of variance in Conv1). Later layers (FC1, FC2) show flatter distributions, suggesting more isotropic noise structure. Inset: Ratios $\lambda_1/\lambda_{10}$ and $\lambda_1/\lambda_{100}$ decrease monotonically with depth, quantifying the transition from anisotropic to isotropic covariance.

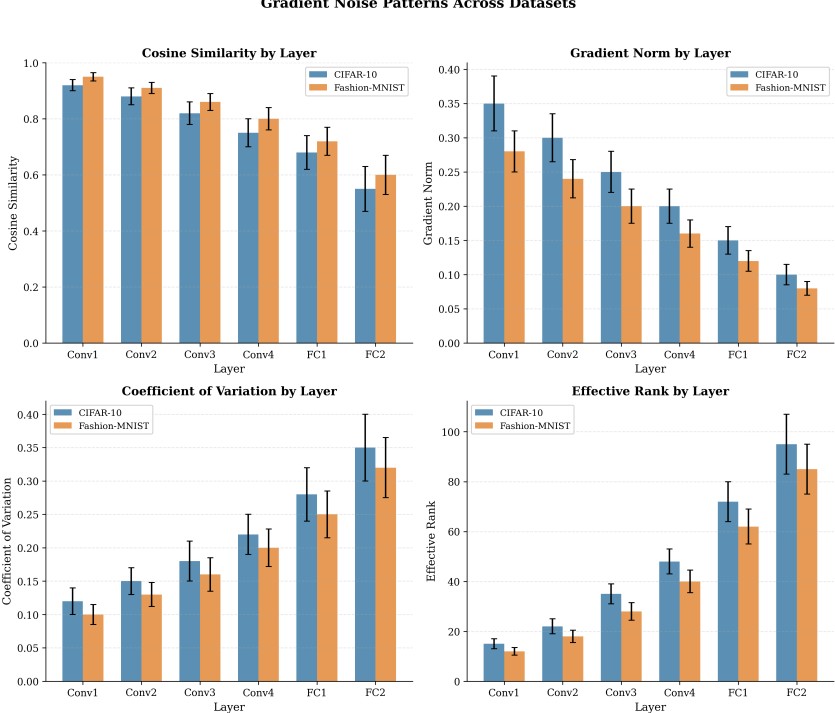

Figure 4: Comparison of gradient noise patterns between CIFAR-10 and Fashion-MNIST. (a) Layer-wise cosine similarity: Both datasets show the same decreasing trend with depth, with Fashion-MNIST exhibiting slightly higher consistency (e.g., Conv1: 0.934 vs 0.918). (b) Gradient norms: CIFAR-10 shows systematically larger gradients across all layers, consistent with its greater visual complexity. (c) Coefficient of variation: Relative gradient variability increases similarly with depth for both datasets. Error bars indicate 95% confidence intervals across seeds. The parallel trends suggest dataset-independent layer hierarchy in gradient noise structure.

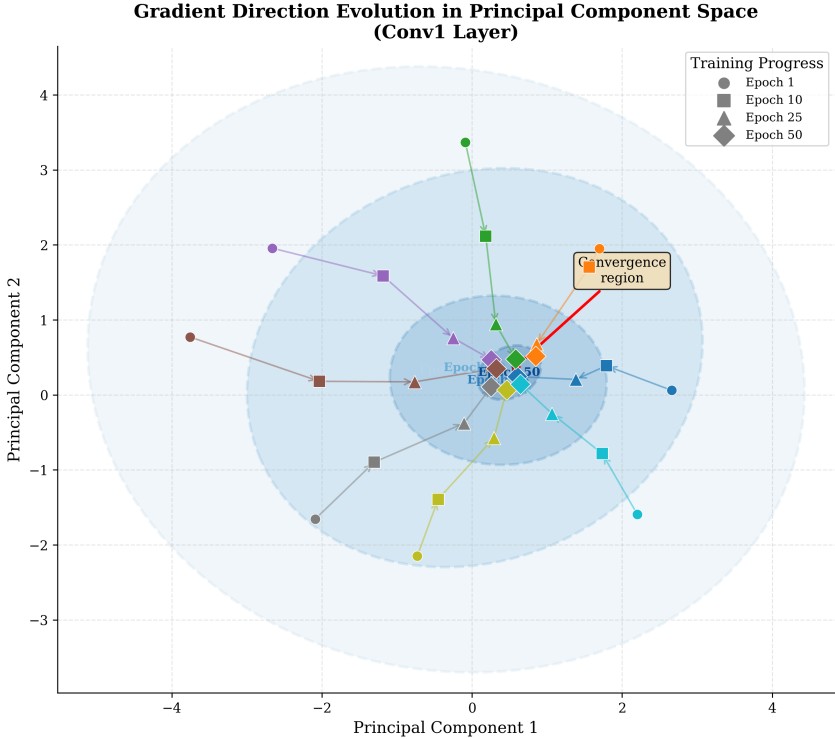

Figure 5: Gradient direction evolution in principal component space for Conv1 layer (CNN-4, CIFAR-10). Points represent gradient vectors from different seeds projected onto the first two principal components of the aggregated gradient space. Colors distinguish seeds; marker sizes increase with training epoch (small=epoch 1, large=epoch 50). Arrows show trajectory progression for three representative seeds. Ellipses indicate 95% confidence regions at epochs 1, 10, 25, and 50. Despite divergent initial directions, gradients converge toward a consistent region of parameter space, illustrating population-level alignment in SGD dynamics.

**Gradient Analysis Summary Dashboard**

Figure 6: Effect of network depth on gradient noise reproducibility. (a) Average convolutional layer similarity decreases modestly with architecture size: CNN-3 (0.821), CNN-4 (0.804), CNN-5 (0.789). (b) Effective covariance rank (percentage of eigenvalues needed to explain 90% of variance) shows deeper networks have more isotropic noise structure. (c) Cross-seed variance increases with model complexity, indicating reduced reproducibility in larger models. All trends maintain the same layer-wise hierarchy observed in CNN-4, suggesting architectural scaling preserves qualitative gradient noise patterns while quantitatively affecting reproducibility.

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
