# OpenReview forum: "LAYER-DEPENDENT STRUCTURE IN GRADIENT NOISE OF SMALL CONVOLUTIONAL NETWORKS"
_ICLR.cc/2026/Workshop/Sci4DL — Sci4DL 2026_

### Official Review · Reviewer_8jm5 · 2026-02-26

**Fit:** 2
**Significance:** 2
**Confidence:** 2

**Summary:**

The paper studies the statistical structure of gradient updates to ask: do independently trained networks produce reproducible gradient patterns across random seeds? The authors study small CNNs on CIFAR10 and Fashion-MNIST, finding that early layers have more directional consistency across seeds than deeper and FC layers. The implications are that SGD gradient noise exhibits systematic structure.

**Strengths:**

* The topic is interesting and worth exploring.
* Clear results and analysis.
* Promising for more studies and insights into deep learning gradient dynamics.

**Suggestions:**

Major Comments
* Include results for optimizer study, can go in the appendix.
* Clarify what the columns "Early Epochs" and  "Late Epochs" refer to in Table 1.

Minor Comments
* The appendix Figure 2 color bar is off (located inside the right-most plot).
* The reference section should come before the appendix section.
* The methodology section is blocky and would benefit from paragraph splits, such as architecture, training, and gradient analysis.
* In the appendix, it would be good to include the accuracy + std from the 20 independent training runs for each architecture as a sanity check.
* In sections 4.1 and 4.2, do the same patterns hold for CNN-3 and CNN-5?
* Specify if Table 3, Figure 3, and Figure 4 results use the CNN-4 architecture.
* The labels in the center of Figure 5 are hard to read due to overlap.
* The caption of Figure 6 does not seem to align with the subfigures. You should have a sentence in the caption for each panel A-F with the correct description.
* The Appendix section should be expanded to include references to each figure and descriptions of the plots and results.

---

### Official Review · Reviewer_z6aB · 2026-02-26

**Fit:** 3
**Significance:** 2
**Confidence:** 2

**Summary:**

This paper empirically studies statistics of gradient noise throughout training for different training runs, with randomness present at initialization and at mini-batch sampling. The authors record a number of different measures, such as norm and covariance structure. They discuss how the the statistics change throughout training and across the depth of the model. Directions of future work are discussed.

**Strengths:**

- Interesting analysis of the structure of SGD noise during training of convolutional architectures.

**Suggestions:**

- I would recommend expanding Section 4.4.2 that discusses the effect of the optimiser on the conclusions. As far as I can tell, the results of that section are not presented in the paper, but only described. Without a careful comparison in such a section, this experimental study lacks actionable conclusions. On the other hand, an interesting case study of how the gradient statistics change as the optimizer changes could be interesting to a lot of people in the community.

---

### Official Review · Reviewer_obrh · 2026-02-27

**Fit:** 2
**Significance:** 2
**Confidence:** 2

**Summary:**

The work studies gradient statistics for small CNNs across random initialization and mini-batch sampling on two benchmark image datasets. The authors find that these gradients exhibit a systematic, reproducible structure across random seeds, with early-layer gradients being more aligned and exhibiting more skewed distributions than those in deeper layers.

**Strengths:**

The analysis is potentially interesting and informative on training dynamics.

**Suggestions:**

The analysis should be extended to optimizer choices and larger architectures, to check if the findings are consistent.
The presentation can be improved: some metrics are not clearly associated with a moment in training (initialization, end of training, etc.), or if the average is done across the dynamics. Also, the labels "Early/Late Epochs" are vague. In Fig. 2 the colorbar is hidden in the plot. I could not associate Fig.6 to its caption.

To further improve the analysis, it would be interesting to understand the source of alignment in early layers: how much proximity to the input contributes to the alignment? Do the statistics in the gradient reflect the statistics in the data?

---

### Meta-Review · Area_Chair_Xkiq · 2026-03-01

**Recommendation:** Accept

**Metareview:**

Reviewers find the topic and results interesting, although further studies are suggested to understand the causes of alignment in the shallow layers and divergence in the later layers.

---

### Decision · Program_Chairs · 2026-03-02

Accept